# Host Combats IBDV Infection at Both Protein and RNA Levels

**DOI:** 10.3390/v14102309

**Published:** 2022-10-21

**Authors:** Shujun Zhang, Shijun Zheng

**Affiliations:** 1Key Laboratory of Animal Epidemiology of the Ministry of Agriculture, College of Veterinary Medicine, China Agricultural University, Beijing 100193, China; 2College of Veterinary Medicine, China Agricultural University, Beijing 100193, China

**Keywords:** infectious bursal disease virus, host response, viral infection, antidefense

## Abstract

Infectious bursal disease (IBD) is an acute, highly contagious, and immunosuppressive avian disease caused by infectious bursal disease virus (IBDV). In recent years, with the emergence of IBDV variants and recombinant strains, IBDV still threatens the poultry industry worldwide. It seems that the battle between host and IBDV will never end. Thus, it is urgent to develop a more comprehensive and effective strategy for the control of this disease. A better understanding of the mechanisms underlying virus–host interactions would be of help in the development of novel vaccines. Recently, much progress has been made in the understanding of the host response against IBDV infection. If the battle between host and IBDV at the protein level is considered the front line, at the RNA level, it can be taken as a hidden line. The host combats IBDV infection at both the front and hidden lines. Therefore, this review focuses on our current understanding of the host response to IBDV infection at both the protein and RNA levels.

## 1. Introduction

Infectious bursal disease (IBD), also known as Gumboro disease, is an acute, highly contagious, and immunosuppressive avian disease caused by infectious bursal disease virus (IBDV). It was first described in the 1960s in Gumboro, Delaware, United State [1,2]. Since then, IBDV has been widely distributed around the world [3,4]. IBDV infects chickens, especially chicks at the age of 3–6 weeks at the final stage of bursa of Fabricius (BF) development [5]; it can directly attack and destroy the BF, the central immune organ of chickens for the development and maturation of B lymphocytes [6]. Furthermore, IBDV infection induces apoptosis in B lymphocytes not only in the BF but also in the spleen and peripheral blood [7], leading to immunosuppression in surviving chickens, followed by increased susceptibility to other pathogenic infections and vaccination failures [8,9]. The factors that influence IBDV-induced mortality are complicated, involving different strains, levels of virulence of the virus, doses of infection, ages, and breeds of chicken, and levels of passive immunity [10]. The mortality of chickens infected with variant IBDV (vIBDV) and very virulent IBDV (vvIBDV) may reach up to 30–100% [11,12,13], leading to huge economic losses to the poultry industry worldwide. Although the vaccination of chickens with live attenuated vaccines or inactivated vaccines is efficacious for the control of IBD in some regions, outbreaks of IBD are still frequently observed due to the emergence of vIBDV strains and new recombinants in the field [14,15,16]. A recent study comparing the pathogenicity of three selected isolates belonging to different genotypes in chickens that had recovered from IBD outbreaks during 2020–2021 revealed that the typical vvIBDV isolate, BD-25 (A3B2), the most virulent strain, caused the disease; it had 100% morbidity and 90% mortality, while the segment-reassortant vvIBDV isolate BD-28 (A3B3) had 50% morbidity and 30% mortality. However, the gross and histopathological lesions of the two vvIBDV strains in the bursa were similar, suggesting that the differences in the morbidity and mortality caused by the two vvIBDV isolates may be due to their genetic composition [17]. Similar to the earlier study [18], the classical virulent isolate BD-26 (A1aB1) did not cause any clinical disease [17]. These results also suggest that three genotypes of IBDV co-spread in chicken flocks in Bangladesh. In addition, it was reported that most IBDV strains circulating recently in North America and Asia were variant strains of IBDV that mainly resulted in subclinical infections, causing enormous economic loss because of serious immunosuppression [19,20]. Importantly, in at least 11 provinces in China, novel variant IBDV strains were isolated from chickens vaccinated against vvIBDV, suggesting that variant IBDV strains could break through the immunoprotection induced by anti-vvIBDV vaccines [20,21]. Furthermore, live attenuated vaccines may induce immunosuppression in chickens [22]. Thus, a complete understanding of the mechanisms underlying IBDV–host interactions will be of great help in the development of novel vaccines.

Viruses replicate and orchestrate the assembly of viral components in host cells to produce a progeny virus, which can affect host cell synthesis, metabolism, and other normal physiological functions, eventually causing diseases in the host [23]. In order to combat pathogens, the host gradually evolves to form an integrated and complicated antiviral immune response. Innate immunity is the first line of host defense against invading microorganisms [24]. Recognition of the invading pathogen by the host is the initial and most critical step for the elicitation of the innate immune response; it relies on the engagement of pathogen-associated molecular patterns (PAMPs) by the host pattern recognition receptors (PRRs), such as Toll-like receptors (TLRs), nucleotide-binding oligomerization domain (NOD)-like receptors (NLRs), RIG-I-like receptors (RLRs), etc. [25,26], and initiation of immune signaling to induce the host response against pathogenic infection [27]. Subsequently, the adaptive immune response is elicited to facilitate the clearance of pathogens and the generation of immunological memory [28]. Meanwhile, pathogens have also developed multiple mechanisms to escape the host immune response [29]. Thus, the war between virus and host always continues. As for the poultry industry, prophylactic strategies using vaccines can, in most cases, prevent and control IBD outbreaks in flocks in epidemic areas. However, because of the constant evolution of the emerging IBDV variants, as well as the lack of guidance for vaccine development, it is difficult to develop highly efficacious vaccines for the prevention and control of IBD [30,31]. Therefore, there is a huge demand for novel and efficient vaccines against IBDV infection. Investigation into cellular factors and/or pathways that directly or indirectly interfere with virus replication would provide valuable clues to the development of novel approaches to the successful control of IBD. The present review focuses on the current understanding of the host response against IBDV infection at both the protein and RNA levels.

## 2. Virus Characteristics

IBDV, a non-enveloped double-stranded RNA virus with a single-shelled icosahedral capsid, belongs to genus *Avibirnavirus* in the family *Birnaviridae* [32]. Its genome consists of bi-segmented RNA encoding only five viral proteins, and each of them may perform multiple functions [33]. Segment A is 3.2 kb in length, and contains two partially overlapping open reading frames (ORFs) encoding a large polyprotein precursor (pVP2-VP4-VP3) and a small nonstructural protein, VP5 (17 kDa) [34]. The polyprotein is later autocatalytically cleaved into three polypeptides, the precursor of the capsid protein pVP2 (54 kDa) [35] that will be further processed into the mature form of VP2 (48.5 kDa) by VP2 itself [36], the scaffold protein VP3 (28 kDa) [37] and the serine protease VP4 (25 kDa) [38]. VP2 is responsible for antigenic variation, serving as a neutralizing antigen [39]. Both VP2 and VP5 play a crucial role in IBDV-induced apoptosis in host cells [40,41,42]. VP3 is an internal capsid protein and can induce group-specific antibodies [43]. Additionally, VP3 binds not only to VP1 and VP2 [44,45], but also to viral genomic dsRNA [46] to inhibit the recognition of viral dsRNA by MDA5, thereby suppressing MDA5-dependent IFN-β production [47]. VP4 can also suppress type I interferon expression by interacting with the glucocorticoid-induced leucine zipper (GILZ) [48,49]. Recently, it was found that VP4 contributes to enhanced virulence of vvIBDV compared with classic IBDV (cIBDV) [50]. Segment B is 2.8 kb in length; it encodes VP1 (90 kDa), an RNA-dependent RNA polymerase protein (RdRp) [51] which is associated with the viral genome, is involved in mRNA synthesis and viral replication [52], and is responsible for the virulence of IBDV [53,54].

## 3. Innate Immune Response of Host to IBDV Infection

The host’s innate immunity forms the first line of antidefense against pathogenic infection. PRRs, expressed in nearly all systemic cells, play a critical role in the initiation of the innate immune response through the recognition of PAMPs, which results in the production of type I and III interferons (IFNs) and other proinflammatory mediators (e.g., cytokines, chemokines, and antimicrobial peptides) [28]. There is growing evidence for the inflammatory response triggered by IBDV infection in chickens [55,56], and it was found that the number of CD4^+^ cells, CD8^+^ cells, and macrophages dramatically increased in the bursa of chickens 1 day after IBDV infection [57], and so did the expression of pro-inflammatory cytokines IL-1β, IL-6 and CXCLi2; however, the expression of anti-inflammatory cytokine TGF-β4 decreased in the bursa of IBDV-infected chickens [58]. Similarly, in the spleen of IBDV-infected chickens, the expression of the mRNA of pro-inflammatory cytokines and chemokines increased [59]. These findings have been recently supported by the upregulated expression of chicken macrophage migration inhibitory factor (chMIF) in primary bursal cells and DT40 cells infected with vvIBDV [60]. Furthermore, chMIF can promote the transcription of proinflammatory cytokines in chicken primary cultured macrophages and induce the migration of peripheral blood mononuclear cells [61], revealing the mechanism of the vvIBDV-mediated initiation of the proinflammatory response. A recent study comparing the transcriptional profiles of White Leghorn chickens of different inbred lines infected with vvIBDV revealed that different White Leghorn lines displayed different disease outcomes following infection with a lower dose of IBDV, and the severity of IBD was associated with significantly enhanced inflammation mediated by chemokines and cytokines, cytoskeletal regulation by Rho GTPases, nicotinic acetylcholine receptor signaling, and Wnt signaling in the BF [62]. It was found that the transcription levels of IL-1β, IFN-β, caspase-1, and NLRP3 increased in IBDV-infected DF-1 cells, and the knockdown of NLRP3 enhanced viral loads, indicating that IBDV infection activates NLRP3 inflammasome in DF-1 cells [63]. However, this study was short of necessary data regarding ASC, caspase 1 or 11, IL-1, IL-18, and cleaved gasdermin at the protein level, which are critical to determining the formation of inflammasome and the induction of subsequent pyroptosis. Nowadays, solid experimental evidence is still required to confirm the IBDV-induced formation of inflammasome, as well as pyroptosis. In principle, the innate immunity of chickens serves as the first line of host defense against IBDV infection, including the inflammatory response, which, in a normal range, promotes phagocytosis and the clearance of IBDV. However, if vvIBDV infection induces an acute and over-inflammatory reaction completely beyond control, the excessive inflammation might result in a ‘cytokine storm’, leading to severe consequences such as sepsis or even death, and the damaged BF will be hardly recovered in the surviving individuals [64]. There is no doubt that the inflammatory response in chickens with IBDV infection is closely associated with the severity of the disease, as well as BF damage.

IFNs are the most important anti-IBDV factors in the host response [65]. The interferon-stimulated genes (ISGs) induced by type I IFNs can affect the virus life cycle at different stages, such as cell entry, replication, transcription, assembly, and release [66]. Because the RIG-I is genetically absent in chickens, the IBDV genome, dsRNA, is mainly recognized through MDA5 and TLR3 [67,68]. In chickens infected with DT40 cell-derived vvIBDV, the expression of TLR1, TLR2, TLR4, TLR3, and TLR5 in the bursa increased [69]. Similar results indicated that the mRNA level of chTLR3 increased in chickens infected with the IBDV field strain NN040124 in PBMCs and DF-1 cells, as well as in bursal tissue, but its downstream effector, IFN-β, decreased at an early stage of infection; this suggests that IBDV evolved with varied strategies to inhibit the production of type I IFNs to survive in the host [70]. In contrast to IBDV-infected HD11 cells, mRNA expression in both TLR3 and TLR7 (which recognize single-stranded viral RNA) in the bursa decreased. The differences in TLR expression between different studies might be due to the different strains of IBDV, because it was found that the expression of TLR3 was upregulated after classical IBDV infection but downregulated after variant IBDV infection [59,71], suggesting that the different expression of TLRs in IBDV-infected cells might be related to the severity of the diseases caused by different strains of the virus.

IBDV infection induced dose-dependent upregulation of chicken MDA5 and IFN-β in chicken HD11 cells, and knockdown of chMDA5 expression led to higher IBDV RNA contents and reduced expression of IRF-3 and IFN-β [72]. It was found that the mRNA expression of TLR1LB, TLR2A, TLR3, TLR4, TLR15 (and TLR21 in the spleen), and MDA5 was upregulated in the bursa early during IBDV infection, whereas the expression of TLR1LA, TLR2B, and TLR7 was downregulated; meanwhile the expression of the antiviral factors IFN-α and IFN-γ and the interferon (IFN)-induced transmembrane proteins (IFITM) 1, IFITM3, and IFITM5 was upregulated in the bursa of IBDV-infected chickens, suggesting that IBDV infection triggered the host innate immune response [73]. These findings revealed that both TLR3 (which sensed viral dsRNA in the endosome) and MDA5 (which senses cytosolic viral dsRNA) play critical roles in sensing the IBDV genome, subsequently initiating the immune signaling pathways of the host in response to IBDV infections.

Defensins are important small antimicrobial peptides in innate immunity, exerting a direct antimicrobial effect on pathogens (bacteria, fungi, protozoa, and enveloped viruses) [74], and they also play a role in immunomodulation [75]. It was found that chickens immunized with the IBDV VP2 gene, together with the chicken β-defensin-1 (AvBD1) gene, had greater antibody levels than those immunized with the IBDV VP2 gene alone, indicating that AvBD1 had an adjuvant effect on promoting the efficacy of the DNA vaccine [76]. It was reported that chicken intestinal antimicrobial peptides (CIAMP) could increase the antibody titers of chickens immunized with the IBDV vaccine, suggesting that CIAMP could modulate the humoral immune response to IBDV infection [77]. It seems that the different genetic backgrounds of chickens cause some differences in the innate immune response to IBDV infection, and different strains of IBDV initiated varying degrees of innate immunity [50,78]. This might give us a chance to consider the possibility of the development of a breeding-selection strategy for IBDV-resistant breeder chickens. The more we know about the influence of chicken genetic backgrounds on innate immunity against IBDV infection, the more effective strategies we can make for the control of IBDV infection.

## 4. Adaptive Immune Response of Host to IBDV Infection

### 4.1. Humoral Immune Response of Chicken to IBDV Infection

The adaptive immunity of the host is responsible for the specific recognition and elimination of pathogens during the later phase of pathogenic infection after innate immunity has failed to clear pathogens completely in the early phase of the disease [79]. The humoral immune response is mainly carried out by B lymphocytes. Unfortunately, the immature B cells in the BF of chickens are target cells of IBDV infection [80], and IBDV-induced apoptosis in B lymphocytes directly results in the severe depletion of B cells in the BF [81], leading to the destruction of the chicken’s immune system. Consistent with previous studies [82,83,84], a recent publication shows that the IBDV strain LJ-5, a new isolate of very virulent IBDV (vvIBDV), decreased the bursa index, B lymphocyte viability, and immunoglobulin (Ig) levels, including IgM and IgA in the bursa and IgY in the sera [85]. Using single-cell RNA sequencing (scRNA-seq) and flow cytometry, it was found that the B cell population in the bursa of IBDV-infected chickens sharply decreased compared with that of controls, and similarly, IBDV infection reduced the number of IgM^+^ and IgY^+^ B cells in the bursa. Furthermore, it was found that IgY^+^ and IgA^+^ B cells were more abundant than IgM^+^ B cells, and that BLMP1 and IRF4, which are responsible for the switching of IgA, were highly expressed in IBDV-infected chickens; this indicates that IBDV infection facilitated IgA secretion in the bursa, and IgY^+^ and IgA^+^ cells might be responsible for the production of antibodies against IBDV [86]. Despite the depletion of immature B cells in the bursa of chickens due to IBDV infection, the mature specific B cells will expand after contact with IBDV and elicit a strong humoral immune response in chickens recovered from the acute phase of IBD [87]. In a recent study, the pathogenicity of the Malaysian variant of IBDV and vvIBDV was comparatively evaluated in specific-pathogen-free (SPF) chickens based on gross and histopathological examinations and viral load, and it was found that even though the two strains of IBDV differed in their viral loads, virulence, and persistence, both strains could elicit a substantial antibody response 7 days after IBDV infection; this suggests that the humoral immune response may play a critical role in the control of IBDV infection [88].

### 4.2. Cell-Mediated Immune Response of Chicken to IBDV Infection

The cell-mediated immune response is mainly carried out by T lymphocytes. The increased number of T lymphocytes and the production of related cytokines and chemokines in the bursa of chickens infected with IBDV suggests that the cell-mediated immune response could be elicited by IBDV infection [89]. It was reported that the expression of Th1 cytokines (IFN-γ, IL-2, and IL-12P40) increased in the bursa of chickens after infection with vvIBDV, but in chickens infected with the IBDV Ts strain, a cell-adapted virus, the expression of Th2 cytokines (IL-4, IL-5, IL-13, and IL-10) was much higher than that of Th1 cytokines; this suggests that the cell-adapted strain of IBDV mainly induced the humoral immune response [90]. Interestingly, it was found that in chickens treated with thymectomy (Tx) and Cyclosporin A (CsA), a drug that selectively suppresses T cell function by inhibiting expression of the IL-2 receptor and by blocking IL-2-mediated signal transduction [91], the IBDV-antigen load in the bursa was significantly greater than that of T cell-intact chickens; this indicates that the functional T cells played an important role in the control of virus replication; however, apoptosis in Tx-CsA birds was reduced compared to that in T cell-intact chickens [92], suggesting that bursa cell destruction induced by IBDV infection may be mediated by cytotoxic T lymphocytes (CTL). Meanwhile, the number of repopulating bursa follicles increased in Tx-CsA-chickens compared to that in T cell-intact chickens, indicating that the presence of functional T cells delayed recovery from the IBDV-induced depletion of bursal follicles [92]. Furthermore, the infection of chickens with IBDV increased the number of CD8^+^ T cells in the bursa and spleen, and the expression of Fas and Fas ligand (FasL), perforin (PFN), and granzyme-A (Gzm-A) in the bursa and spleen was significantly upregulated; this suggests that CTL may be involved in the clearance of IBDV from the target organ, the bursa, and peripheral tissues through both the Fas–FasL and perforin–granzyme pathways [93]. Thus far, the exact mechanism by which T-cells respond to IBDV infection is still unclear. It is very important to determine whether IBDV carries dominant protective epitopes for T-cell activation. The activated T lymphocytes will recognize the epitope presented by avian MHC-I (called B in poultry) of the IBDV-infected cells for the induction of cell death, followed by the release of the IBDV antigen from destructed cells so that the free viral antigens can bind to specific antibodies to form an antigen–antibody complex. Subsequently, they are phagocytosed by macrophages via Fc receptor-mediated opsonization or direct macropinocytosis. Thus, T cell-mediated immunity is crucial to the clearance of IBDV by the immune system from the infected host.

## 5. Host Response to IBDV Infection at the Protein Level

A better understanding of molecular mechanisms for virus infection will be very helpful in the development of novel vaccines or antiviral drugs, and further investigation into host–pathogen interactions should be encouraged to elucidate the molecular mechanisms underlying the host response to viral infection. Recently, reports on extensive and complex interactions between the virus and host at the molecular level have prompted researchers to the study of virus–host interaction network and the proposal of “virus–host interactomics” [94]. Targeting cellular factors for antivirus therapy will be a new approach that could overcome viral mutations and resistance. In recent years, remarkable progress has been made in IBDV–host interactions, which facilitates the search for cellular targets that could inhibit IBDV replication, providing a theoretical basis for the development of novel vaccines or antiviral drugs.

Control of the amplification of the viral genome is an effective strategy of the host to inhibit IBDV replication. The translational eukaryotic initiation factor 4AII (eIF4AII), a host cellular factor participating in the translation initiation of most cellular mRNA [95], interacted with VP1 to inhibit viral RNA polymerase activity in IBDV-infected DF-1 cells, resulting in a decrease in IBDV replication (Figure 1), indicating the suppressive role of host factor eIF4AII in the replication of a dsRNA virus [96]. Additionally, as shown in Figure 1, another cellular factor, nuclear factor 45 (NF45), which participates in RNA export and mediating mRNA stability and translation, was reported to specifically colocalize with VP1, VP2, and VP3 in IBDV-infected DF-1 cells, interacting with the viral replication complexes and suppressing IBDV replication [97]. Cyclophilin A (CyPA), a ubiquitously expressed protein possessing peptidyl-prolyl cis–trans isomerase (PPIase) activity, plays an important role in protein modification, folding, trafficking, and transcription regulation [98,99]. CyPA is also involved in viral infection through various mechanisms [100]. It was reported that in IBDV-infected cells, CyPA interacted with VP4 to inhibit IBDV replication (Figure 1), suggesting that CyPA affects the enzymatic activity of VP4 and/or activates the innate immune response against IBDV infection; however, further evidence is still required to determine its role [101]. Similarly, in a recent study, RNA-seq was used to investigate host factors involved in IBDV infection; it was found that TRIM25 interacted with VP3 and mediated its ubiquitination and subsequent degradation, thereby restricting IBDV replication (Figure 1). These findings suggest that TRIM25 is a host factor restricting viral replication [102].

Autophagy is a highly conserved cytoplasmic pathway of maintaining physiological stability in eukaryotes, which sequesters and removes unwanted self-materials by degrading them into amino acids for re-use. A growing number of studies established an antiviral role of autophagy in the host response against pathogenic infection [103,104]. It was reported that the interaction of IBDV VP2 with heat shock protein 90 (HSP90AA1) induced autophagy through the HSP90AA1-AKT-mTOR pathway early during infection, and the activated autophagy inhibited virus replication (Figure 1), suggesting that autophagy is involved in host defense against IBDV infection [105]. Recently, it was found that autophagy cargo receptor p62 interacted with IBDV VP2, which enhances the induction of autophagy and promotes the autophagic degradation of VP2 (Figure 1), suggesting that the p62-mediated autophagic degradation of VP2 may play a role in the presentation of the VP2 peptide by MHC to initiate the adaptive immune response [106]. Of note, the autophagy that degrades foreign materials (such as pathogens) is often called xenophagy. In a recent study, the autophagy cargo receptor SQSTM1 was found to directly bind to IBDV dsRNA through the amino acid sites R139 and K141 and mediated the autophagic degradation of viral RNA, thereby suppressing IBDV replication (Figure 1) [107]. These findings suggest that the selective autophagy of the viral genome by the host (xenophagy) plays a critical role in the control of IBDV infection.

## 6. Host Response to IBDV Infection at the RNA Level

It is generally accepted that non-coding RNAs (ncRNAs) are a kind of RNA that does not encode for proteins, and it has been reported that 75% of the human genome is transcribed into RNA, while only 3% is transcribed into protein-coding mRNAs [108]. Thus, the ncRNAs abounding in the body play crucial roles in various biological processes, including cancers and other types of diseases [109,110]. According to their length, structure, and location, ncRNAs have been divided into several classes, mainly including microRNAs (miRNAs), long ncRNAs (lncRNAs), and circular RNAs (circRNAs) [111], but there are still many ncRNAs that need to be further characterized. There have been many pieces of evidence that ncRNAs participate in the host response to virus infection, especially based on the results from those studies on miRNAs [112,113]. In the host response to IBDV infection, both cellular proteins and ncRNAs participate in the antidefense process. If the interaction of the host with IBDV at the protein level is imagined as the front line of a battle between invaders (pathogens) and defenders (host cells) [114], their fight at the RNA level can be seen as the hidden front line. The following are some examples indicating the IBDV–host interaction at the RNA level.

### 6.1. The Antiviral Role of miRNAs in Host Response to IBDV Infection

miRNA, approximately 22 nucleotides (nt) in length, is the most widely studied ncRNA that regulates the expression of target mRNAs [115]. In IBDV-stimulated chicken dendritic cells (DCs), 991 conserved miRNAs were detected [116]. Among them, there were 18 miRNAs whose expression was significantly altered, including 11 with downregulated expression and 7 with upregulated expression. Using the KEGG pathway and the BIOCARTA database for the analysis of predicted targets, it was found that MAPK signaling, P38 signaling, and the transcription factor CREB and its extracellular signals were involved in the host response to IBDV infection [116]. In a recent study, it was found that a total of 1710 miRNAs were identified in vvIBDV-infected BF tissues and mock-infected controls, and there were 42 miRNAs with upregulated expression and 35 miRNAs with downregulated expression. miR-1684b-3p, gga-miR-1788-3p, and gga-miR-3530-5p had especially significantly varied expression levels following IBDV infection, with their potential target genes, including THBS1, STAT1, STAT3, and MYD88 [117]. These predicted results suggest that these gga-miRs might play a role in apoptosis, inflammation, and the IFN response, but further evidence will be required to determine the exact role of these miRNAs in the host response to IBDV infection.

Lines of evidence indicate that gga-miRNAs exert antiviral effects on IBDV infection, such as gga-miR-454 [118], gga-miR-130b [119], gga-miR-155 [120], gga-miR-21 [121], and gga-miR-27b [122]. It was reported that 296 miRNAs were differently expressed in DF-1 cells with IBDV infection. MiR-130b, an upregulated miRNA in IBDV-infected cells, was found to inhibit IBDV replication by directly targeting IBDV segment A and cellular SOCS5 [119]. SOCS5 belongs to the SOCS (suppressor of cytokine signaling) family and negatively regulates JAK-STAT signaling. Moreover, it plays an important role in the host response to viral infections by directly binding to the JAK kinase domain via a region in the N terminus, inhibiting the autophosphorylation of JAK1 and JAK2 [123,124,125]. Gga-miR-130b caused a decrease in SOCS5 expression, followed by enhancement of the mRNA expression of STAT1, 3, and 6, as well as phosphorylation of STAT1, subsequently activating the type I IFN signaling pathway, which plays a major role in the host antiviral response (Figure 2). In addition to IBDV, miR-130b also acts as an antiviral factor against other viruses [126], suggesting the potential of miR-130b as a novel therapeutic target to treat viral diseases [119]. Similarly, it was reported that gga-miR-454 markedly decreased in IBDV-infected DF-1 cells, and the overexpression of miR-454 greatly enhanced the expression of IFN-β, as well as that of IRF3 and p65, thereby inhibiting IBDV replication [118]. Interestingly, in this study, it was found that miR-454—by targeting SOCS6, a negative regulator of JAK-STAT pathway—inhibited IBDV replication by enhancing the type I IFN response. In addition to targeting cellular SOCS6, miR-454 could also directly target IBDV segment B to inhibit the transcription of the viral genome (Figure 2). Furthermore, the binding site of gga-miR-454 in the genome of vvIBDV was found to contain mutations, whereas it is relatively conserved in most classical and low-virulence IBDV strains. This unexpected finding suggests that vvIBDV strains might evolve, with mutations in its genomic binding site targeted by cellular miRNAs to evade the immune response of the host for its survival; it also indicates a critical role for miR-454 in host defense against IBDV infection [118]. However, the mechanisms underlying the IBDV-induced inhibition of gga-miR-454 expression remain unclear.

Gga-miR-155 is one of the most extensively studied miRNAs in the host response to pathogenic infection, playing an antiviral role in host defense [127,128,129]. It was reported that gga-miR-155 could suppress IBDV replication by enhancing type I IFN signaling via the targeting of SOCS1 and TANK, both of which were negative regulators in the innate immune response (Figure 2); this suggests that miR-155 is an important antiviral factor in the host immune response to IBDV infection [120]. A recent publication by our laboratory revealed the molecular mechanisms underlying the IBDV-induced upregulation of gga-miR-155 expression. It was found that upon IBDV infection, the expression of GATA-binding protein (GATA) 3 in DF-1 cells markedly increased; this was attributable to the recognition of viral dsRNA by cellular MDA5, subsequently initiating TBK1-IRF7 signaling pathways, thereby promoting GATA3 expression (Figure 2). Phosphorylated GATA3 translocated to the nucleus where it directly bound to the gga-miR-155 promoter, regulating the expression of gga-miR-155 in IBDV-infected or poly (I:C)-treated DF-1 cells [130]. GATA3 belongs to the GATA-binding protein family, which plays a crucial role in physiological and pathological processes by directly initiating or restricting the transcription of target genes; GATA3, especially, serves as a key transcription factor driving the differentiation of Th2 cells [131,132]. Furthermore, our data show that GATA3 inhibited IBDV replication by promoting the expression gga-miR-155, suppressing IBDV replication by enhancing type I IFN expression (Figure 2). These findings indicate that the transcription factor GATA3 plays a critical role in the host antiviral response by promoting gga-miR-155 expression, which deciphers how the expression of a specific miRNA is regulated in host cells with pathogenic infection; moreover, they remarkably further our understanding of the mechanism of the host response against IBDV infection at the RNA level [130].

It was found that gga-miR-21 increased in IBDV-infected chicken cells, and gga-miR-21 inhibited IBDV replication by directly targeting the 22nt region of VP1 mRNA to block its translation (Figure 2). Furthermore, this 22nt sequence of VP1 was conserved among IBDV strains, indicating that this short sequence is important for the RNA-dependent RNA polymerase function of VP1 in IBDV replication, which may be used as a potential target for new antiviral drug development [121]. Similar to the miRNAs described above, IBDV infection increased gga-miR-27b expression in DF-1 cells by demethylating the pre-miR-27b promoter region, and miR-27b could suppress IBDV replication by upregulating the expression of IFN-β, IRF3, and NF-κB via the targeting of SOCS3 and 6 (Figure 2), the negative regulators of JAK-STAT signaling [122]. In addition to chickens, miR-27b also plays a crucial role in the host immune response against viral infection in mammals [133,134]. For example, recently, it was found that the antiviral role of miR-27b can suppress transmissible gastroenteritis virus (TGEV) replication by directly targeting porcine SOCS6, while miR-27b expression was down-regulated by TGEV infection through the activation of ER stress, thereby splicing the mRNA encoding a potent transcription factor, Xbp1, that could inhibit the activation of the miR-27b promoter; this suggests that TGEV evolved a strategy to avoid the antiviral effect of miR-27b. It is worth noting that miR-27b can also inhibit porcine epidemic diarrhea virus (PEDV) and porcine rotavirus (PoRV) replication [135]. Thus, miR-27b may serve as a broad-spectrum antiviral factor in the host response against viral infection.

### 6.2. The Antiviral Role of lncRNAs in Host Response to IBDV Infection

LncRNAs, >200 nt in length, are dispersed throughout the genome, and are typically expressed at low levels and in a tissue-specific manner, playing an important role in many cellular processes, such as epigenetic control, transcription regulation, translation, RNA metabolism, autophagy, apoptosis, and embryonic development [136,137]. The subcellular localization of lncRNAs is critical to their function. In the cytoplasm, lncRNAs can regulate the activity and levels of miRNAs by sequestering them [137]. Unlike most mRNAs keeping sequences conserved among species, lncRNAs are generally poorly conserved [138], making it more challenging to assess lncRNAs’ function. In many diseases, especially in different types of cancer, the widespread dysregulation of lncRNA expression levels highlights the need to understand the functions and mechanisms of lncRNAs [139,140]. Recently, lines of evidence suggest that lncRNAs play a role in viral infection [141,142,143]. In IBDV-infected chicken DCs, a total of 3900 lncRNAs were detected; among them, 25 were identified to be upregulated and 79 were downregulated, and then, 222 co-expressed lncRNAs/target genes were identified. Using informatics analytic tools, it was found that those differentially expressed genes might be involved in the cellular response to starvation, the negative regulation of binding, and suppression of the apoptotic process. In the same study, some lncRNAs, such as MYOZ1, TMEM130, and UBE2QL1, were found to be significantly increased in the IBDV-infected group, suggesting that lncRNAs play an essential role in the host response against IBDV infection [116]. Similarly, it was reported that lncRNAs were involved in vvIBDV infection, and that 172 lncRNAs were upregulated while 100 were downregulated in the bursa of chickens infected with vvIBDV. The follow-up experiments showed that the lncRNAs LOC107053928, LOC107054815, LOC107053352, and LOC107053557 regulated immunomodulation-related genes, including STAT1, STAT3, STAT4, TRIM25, and IFIH1, suggesting that these lncRNAs might play a vital role in the antiviral response [144]. Recently, it was found that in IBDV-infected DF-1 cells, 13 lncRNAs were upregulated while 35 lncRNAs downregulated, and that the gene pair of loc107051710 and IRF8 has the closest relationship [145]. IRF8, a transcription factor mainly expressed in myeloid cells, including macrophages, monocytes, and dendritic cells, is required for the development and function of these cells, especially IRF8, which plays a critical role in the antimicrobial defense of macrophages via the activation of effector targets in response to IFNγ/PAMP signaling [146]. The knockdown of loc107051710 significantly increased IBDV replication by decreasing the mRNA expression of IRF8 (Figure 2), type I IFNs (IFN-α and IFN-β), STATs (STAT1 and STAT2), and ISGs (OAS, Mx1, and PKR), suggesting that loc107051710 plays an antiviral role in the host response to viral infection [145]. Until now, only a few studies, such as those described above, had reported on the role of lncRNAs in IBDV infection, and most of the results obtained were merely based on bioinformatics analysis. Thus, more efforts will be required to elucidate the exact mechanisms of the lncRNA–virus interaction.

### 6.3. The Antiviral Role of circRNAs in Host Response to IBDV Infection

CircRNAs, a kind of ncRNA generated from linear precursor mRNAs in a nonclassical splicing square with a covalently closed loop, are highly abundant and evolutionarily conserved in eukaryotes [147]. The biological functions of circRNAs, including the regulation of gene expression and translation, and acting as decoys and transporters or scaffolds by interacting with the RNA binding protein (RBP), were very well reviewed [148]. Importantly, it was found that circRNAs could also act as miRNA sponges to suppress their activity by binding to various miRNAs [149,150]. Lines of evidence indicate that circRNAs play a role in various human disorders, and most of them are correlated with tumor progression or repression [151,152]. In recent years, a growing number of studies reported that circRNAs also participated in virus–host interaction [153,154]. In the area of IBDV infection, there was only one report that showed the differential expression profiles of circRNAs, and it indicated that 63 circRNAs were upregulated and 80 downregulated [144]. Using bioinformatics to construct a circRNA-miRNA-mRNA network, it was found that the circRNAs novel_circ_000574 and novel_circ_001469 significantly increased in the bursa of chickens infected with vvIBDV, and that they potentially targeted genes involved in immune-related genes, including *STAT1* and *IRF7,* by binding to miR-1587-x and miR-4507-y, respectively; this suggests that circRNAs play a role in the host response to IBDV infection [144]. More efforts will be required to investigate the mechanisms underlying the abnormal landscape of circRNAs and how circRNAs exert physiological or pathological roles in the host response to IBDV infection.

In summary, ncRNAs participate in the host response to IBDV infection, and among them, miRNAs have been best studied in this process. MiRNAs play an antiviral role in the host response to IBDV infection, mainly through two means: the first is to directly target at the viral genome to inhibit viral replication (gga-miR-130b→segment A, gga-miR-454→segment B, and gga-miR-21→VP1 mRNA), and the second is to inhibit IBDV replication by targeting the negative regulators of the antiviral immune response in host cells (gga-miR-130b→SOCS5, gga-miR-454→SOCS6, gga-miR-27b→SOCS3 and 6, gga-miR-155→SOCS1, and TANK). However, the exact mechanisms underlying the IBDV-induced differential expression of these miRNAs, except for gga-miR-155, remain unclear and are worth further investigating in the future. As for lncRNAs and circRNAs, most publications merely focused on bioinformatics analysis, and the studies regarding the molecular mechanisms underlying the antiviral role of circRNAs and lncRNAs in the host response to IBDV infection are quite limited and require further efforts. Because of the high abundance and multiple functions of ncRNAs, further comprehensive investigations into their roles in the host response to viral infection would be helpful for the development of a new strategy for the effective control of IBDV infection.

## 7. Conclusions

Although a vaccination program for the prevention and control of chicken diseases has been implemented globally, sporadic outbreaks of IBD still occur, indicating that IBD is still not very well controlled and continues to threaten the poultry industry. Thus, it is urgent to develop more effective and safer vaccines for the clinical control of IBD. A complete understanding of the molecular mechanism of the host response against IBDV infection would be of great help in the development of antiviral strategies. Of note, the development of new adjuvants to improve vaccines’ immunogenicity is an effective alternative to protect chickens against IBDV infection, since chIL-2 and chIL-7 cytokines were effective biological adjuvants that enhanced the immunogenicity of the IBDV DNA vaccine [155]. In the IBDV–host interaction, some cellular proteins play an antiviral role through different mechanisms; for example, eIF4AII, NF45, CyPA, and TRIM25 suppressed IBDV replication by interacting with viral proteins or the replication complex, while HSP90AA1, p62, and SQSTM1 inhibited IBDV replication by interacting with viral proteins or genomes to initiate host autophagy, which subsequently induced their autophagic degradation. Investigation into host restriction factors and their molecular mechanisms during IBDV infection would be indispensable and highly encouraged in the future. Host non-coding RNAs play an important role in host defense against IBDV infection and/or regulate IBDV infection directly or indirectly, which may provide a valuable reference for the development of antiviral therapeutics at the RNA level [156]. For example, miR-130b might serve as an antiviral drug for respiratory disease in chickens since the intranasal delivery of miR-130b mimics to piglets exhibited strong anti-PRRSV activity [126]. The host–IBDV interaction at the RNA level still has many questions that need to be addressed since the studies on non-coding RNAs in IBDV-infected host cells are quite limited. Although it is known that the circRNA-miRNA-mRNA network was demonstrated by bioinformatics, further clarification and confirmation of this network via laboratory studies would be helpful in understanding the precise regulatory mechanism of the host response to IBDV infection. It would also be of great help in the development of effective measures for the control of IBDV infection.

## Figures and Tables

**Figure 1 viruses-14-02309-f001:**
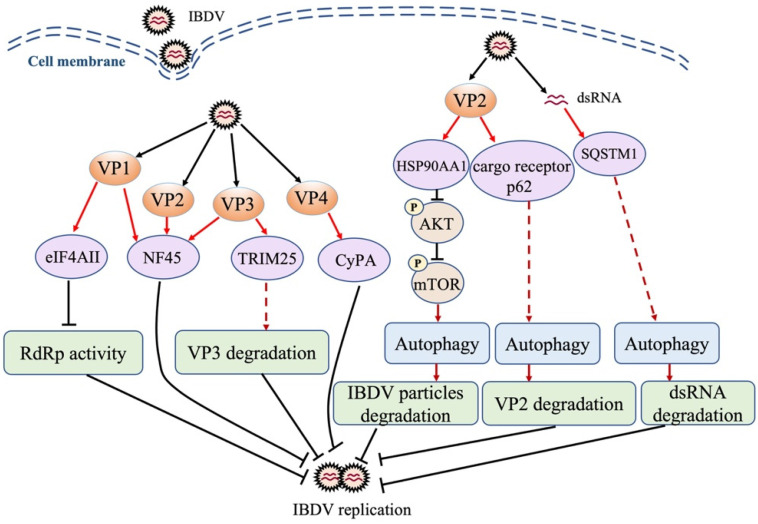
Schematic diagram of the roles of cellular proteins in host response to IBDV infection. IBDV infection triggers host antiviral response, and cellular proteins are involved in this response, inhibiting viral replication via direct interaction with the viral protein or genome. eIF4AII: translational eukaryotic initiation factor 4AII; NF45: nuclear factor 45; TRIM25: tripartite motif 25; CyPA: cyclophilin A; HSP90AA1: heat shock protein 90 AA1; SQSTM1: sequestosome-1; AKT: protein kinase B; mTOR: mammalian target of rapamycin; RdRp: RNA-dependent RNA polymerase protein; dsRNA: double-stranded RNA.

**Figure 2 viruses-14-02309-f002:**
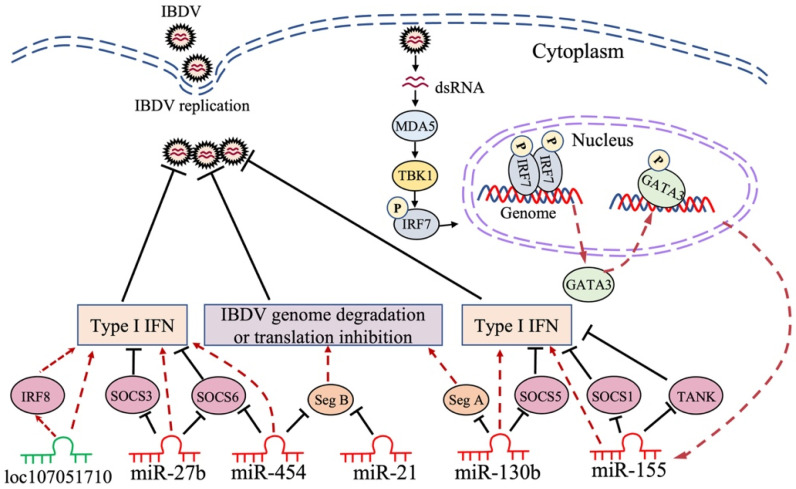
Schematic diagram of the roles of non-coding RNA in host response to IBDV infection. IBDV infection affects the expression of non-coding RNAs in host cells. Non-coding RNAs play roles in inhibiting IBDV replication by directly targeting the viral genome or host gene involved in immune signaling pathways. Seg A: IBDV genome segment A; Seg B: IBDV genome segment B; SOCS1: suppressor of cytokine signaling 1; SOCS3: suppressor of cytokine signaling 3; SOCS5: suppressor of cytokine signaling 5; SOCS6: suppressor of cytokine signaling 6; TANK: TRAF family member-associated NF-κB activator; IRF7: interferon regulatory factor 7; IRF8: interferon regulatory factor 8; GATA3: Gata-binding protein 3; MDA5: melanoma differentiation-associated gene 5; TBK1: TANK-binding kinase 1; IFN: interferon.

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
