# Peer review of "Host Combats IBDV Infection at Both Protein and RNA Levels"

_viruses, 2022, doi:10.3390/v14102309_

Round 1

Reviewer 1 Report

The current review concludes on the host response to IBDV infection at both protein and RNA levels, which will contribute to the development of effective measures for IBDV infection control. However, there is one major issue in the article that requires further discussion. The authors mention that different strains of IBDV initiated varying degrees of innate immunity, humoral immune response and cell-mediated immune response. The prevalence of IBDV in these strains and the relationship between the characteristics of these strains and immunity needs to be further summarized and discussed. In addition, there are some grammatical errors in the manuscript. It requires a native English speaker to further polish the language.

Reviewer 2 Report

This review is well written. There are some minor conerns

1. Line 70, "genus Avibirnavirus in the family Birnaviridae", the genus and family name should be Italicized.

2. Line 125, "such as sepsis", is it means Viremia? 

3. Line 189-190, "it was found that B cells with IgY+ and IgA+ positive were more 189 abundant than IgM+ B cells",  is it means IgY and IgA positive or  IgY+ and IgA+? And at the end of this sentence, the location of B cells should be cleared, in BF or somewhere?

4. Line 339, "the I-IFN" should be "type I IFN" as described.

5. Line 436, "loc107051710" has shown in Figure 2, but no tagging in the text.
